# Possible Points of Ulnar Nerve Entrapment in the Arm and Forearm: An Ultrasound, Anatomical, and Histological Study

**DOI:** 10.3390/diagnostics13071332

**Published:** 2023-04-03

**Authors:** Andrea Ferre-Martinez, Maribel Miguel-Pérez, Ingrid Möller, Sara Ortiz-Miguel, Albert Pérez-Bellmunt, Núria Ruiz, Xavier Sanjuan, Jose Agullo, Juan Ortiz-Sagristà, Carlo Martinoli

**Affiliations:** 1Unit of Human Anatomy and Embryology, Department of Pathology and Experimental Therapeutics, Faculty of Medicine and Health Sciences (Bellvitge Campus), University of Barcelona, 08907 Barcelona, Spain; 2Basic Sciences Department, Universitat Internacional de Catalunya, 08017 Barcelona, Spain; 3Department of Pathology, University Hospital of Bellvitge, 08907 Barcelona, Spain; 4Unit of Pathological Anatomy, Department of Pathology and Experimental Therapeutics, Faculty of Medicine and Health Sciences (Bellvitge Campus), University of Barcelona, 08907 Barcelona, Spain; 5Anesthesiology Department, Fundació Puigvert, 08025 Barcelona, Spain; 6Cattedra di Radiologia “R”-DICMI, Universita di Genova, 16126 Genoa, Italy

**Keywords:** ulnar nerve, nerve entrapment, compressive neuropathy, ultrasound, intermuscular septum, muscular fascia, flexor carpi ulnaris, flexor digitorum superficialis

## Abstract

Background: Ulnar nerve entrapment is one of the most common entrapment neuropathies, usually occurring in the cubital tunnel of the elbow and in Guyon’s canal of the wrist. However, it can also occur at other anatomical locations. Purpose: Our aim was to review other possible locations of ulnar nerve entrapment in an ultrasound and anatomical study. Material and Methods: Eleven upper limbs from eight adult corpses were ultrasonographically examined and subsequently dissected in a dissection laboratory. Four specific anatomical points were analysed, and any anatomical variations were documented. Moreover, six samples of the nerve were taken for histological analysis. Results: Distinct anatomical relationships were observed during ultrasound and dissection between the ulnar nerve and the medial intermuscular septum, the triceps aponeurosis, Osborne’s fascia at the elbow, the arcuate ligament of Osborne and the intermuscular aponeurosis between the flexor carpi ulnaris and the flexor digitorum superficialis muscles. A statistical study showed that these locations are potential areas for ulnar nerve compression. In addition, a fourth head of the triceps brachii muscle was found in some specimens. Conclusion: Results demonstrate that ultrasound is a good tool to investigate ulnar nerve entrapment neuropathy and to identify other anatomical points where the nerve can remain compressed.

## 1. Introduction

Entrapment neuropathies are a heterogeneous group of conditions in which peripheral nerves are chronically compressed or stretched as a result of congenital or acquired disorders or a combination of both [1]. Clinical symptoms depend on the degree and duration of entrapment. They can range from sensory abnormalities (pain, paraesthesia, and numbness) to motor weakness in the area innervated by the affected nerve [2,3,4]. For this reason, accurate clinical evaluations including history taking, physical examinations, and electrodiagnostic tests are the mainstays of diagnosis. However, imaging studies have an important role in identifying the abnormality causing the entrapment neuropathy [2].

As for mononeuropathies, ulnar neuropathy is most often due to compression at different fibro-osseous tunnels locations as the cubital tunnel of the elbow and the Guyon’s canal of the wrist [4,5] or congenital conditions involving accessory muscles [1]. However, ulnar nerve entrapment can occur at other sites in the mid-arm to distal elbow and could have other aetiologies that can increase the risk of nerve compression.

The ulnar nerve (UN) arises from the medial cord of the brachial plexus (C8, T1) and courses along the medial side of the brachial artery in the anterior compartment of the proximal aspect of the arm [2]. It enters the posterior compartment of the arm, crossing the medial intermuscular septum (IMS) from the front to the back [6]. The IMS is a fibrous band that blends with the distal coracobrachialis tendon at the mid-shaft of the humerus and inserts into the medial epicondyle distally [7]. At the level of the elbow, and posteriorly, the UN courses through the epitrochlear tunnel of the elbow, an osteofibrous tunnel composed by the olecranon laterally and the medial epicondyle medially. In fact, the medial epicondyle posteriorly forms a groove through which the ulnar nerve passes and surrounds it partially by its anteromedial side [2,7,8]. The floor of this tunnel is composed of the posterior band of the ulnar collateral ligament [6,8,9,10], while the roof is composed of a fascial sheet also known as Osborne’s fascia [11]. Posteriorly, the nerve enters the hiatus between the ulnar and humeral heads of the flexor carpi ulnaris (FCU) muscle. The two heads of this muscle are connected by an aponeurotic arch, i.e., the arcuate ligament of Osborne [10], which represents a distal expansion of the cubital tunnel retinaculum [11]. Distally, the nerve dives below the FCU to enter the fascial sleeve of the flexor digitorum superficialis (FDS) muscle [12]. It runs between the FDS and the FCU until it reaches the wrist, where it passes through Guyon’s canal and bifurcates into the sensory and motor terminal branches [5,7].

To evaluate the structural aetiology of nerve entrapment, imaging studies can be very useful [2]. Among them, ultrasound (US) is a reliable method for evaluating the anatomy and structures causing nerve compression [13,14]. In fact, US can detect UN instability in a dynamic way [15] as well as asses other anatomical structures [16,17,18].

The aim of this study was to obtain good anatomical knowledge of the UN and its pathway through the arm and forearm, taking into consideration its relationships with other anatomical structures as well as any anatomical variations. This can identify multiple potential sites where the nerve may be compressed, thereby facilitating the diagnosis of entrapment neuropathy by US.

## 2. Material and Methods

This study was conducted on eleven upper limbs (four right upper limbs and seven left upper limbs) obtained from eight adult corpses cryopreserved at −20 °C in the Dissection Laboratory of the Faculty of Medicine and Health Sciences at the University of Barcelona. Among the corpses, six were male (55%), and five were female (45%). The age of the corpses ranged from 65 to 89 years. Specimens were enumerated gradually during the US and anatomical studies. The axillary artery was injected with black latex (Latex Compound Española S.A., Sadabell, Spain) to facilitate its localisation in the US and anatomical studies. Cadavers whose upper limbs were deformed by malformations, traumas, or scars due to surgery history were excluded. Thus, the US study was performed first, followed by the anatomical study, where all the upper limbs were dissected carefully, and then the histological analysis.

### 2.1. Ultrasound Study

The US study was performed with a LOGIQ P6 and P9 ultrasound system (GE ultrasound Korea. LTD. Seongnam, Republic of Korea) and a high-frequency lineal transducer of 6–15 MHz. The UN was studied from its origin at the medial cord of the brachial plexus to its terminal bifurcation at the level of the pisiform bone. First, it was visualised along the transversal axis. Once the nerve was identified, it was followed in the axial images obtained during the alternating cranial and caudal displacement of the transducer. After the assessment in the transverse plane, the nerve was examined in the longitudinal images.

During the movement of the transducer along the transverse plane, changes into the morphology and echogenicity of the nerve as well as the appearance of its surrounding structures were explored. Then, longitudinal US studies were performed to verify any portion of the nerve that was suspicious to be compressed. At all times, the possible presence of anatomical variations was taken into consideration as they could explain UN entrapment.

Four specific locations were evaluated, as they were considered notable points for nerve entrapment (Figure 1): (1) the middle third of the arm, where the nerve crossed into posterior compartment through the IMS; (2) the lower third of the arm, where the nerve is associated with the medial head of the triceps; (3) the level of the elbow, where the path of the nerve is associated with the hiatus between the ulnar and humeral heads of the FCU; and (4) the upper third of the forearm, where the nerve passes through an intermuscular aponeurosis between FCU and FDS.

With the panoramic views of the studied areas obtained with the US transverse plane, an intramuscular needle was used to identify all these anatomical locations to confirm the correct position in the dissection study.

### 2.2. Anatomical Study

An anatomical study was carried out with a careful dissection of the UN along its entire path. At all times, the intramuscular needles placed during US were used as the landmarks of the UN at the specific points mentioned above. First, a longitudinal skin incision at the anteromedial aspect of the arm, from the axilla to the elbow, was made. Then, two terminal horizontal skin incisions were made: the first one at the level of the axilla and the second one at the level of the elbow. The skin was removed, and the entire superficial fascia was dissected. Once the UN was identified at its origin at the medial cord of the brachial plexus, it was dissected thoroughly along its path up to its entrance into the osteofibrous tunnel at the elbow. We paid special attention to its passage through the IMS and to its relationship with the medial head of the triceps brachii in the posterior compartment. The epineurium of the nerve was preserved constantly.

Hereunder, a second longitudinal skin incision at the anteromedial aspect of the forearm, from the elbow to the wrist, and a third horizontal skin incision at the wrist were made. Again, the skin was removed, and the superficial fascia was dissected. The UN was released from the osteofibrous tunnel. Then, it was followed along its path from underneath the aponeurotic arch that joins the two heads of the FCU (with special attention paid to its relationships with the FCU and FDS muscles) to the wrist.

During the dissection of the nerve along its entire course in the arm and forearm, its anatomical relationships with the other structures were studied, while iconographic information (Canon EOS 60D) was recorded. The exact anatomical positions of the intramuscular needles were also verified. Finally, the diameter and perimeter studies of the nerve were performed with a digital calliper and a thin thread, respectively. The four specific anatomical locations studied by US were used as the anatomical landmarks to carry out these measurements. Thus, the diameter and perimeter were measured at (Figure 1):(A)Two centimetres proximal to the IMS, the first anatomical landmark studied by US [1];(B)Two centimetres distal to the IMS, the first anatomical landmark studied by US [1];(C)Three centimetres proximal to the medial epicondyle (ME), the second anatomical landmark studied by US [2];(D)Before the traversing below the hiatus between the two heads of the FCU, the third anatomical landmark studied by US [3];(E)After the piercing of the intermuscular aponeurosis between the FCU and FDS, the fourth anatomical landmark studied by US [4].

### 2.3. Histological Study

From three upper limbs, six samples (2 × 2 cm) of the UN and its surrounding anatomical structures from the anatomical landmarks studied were obtained to carry out a histological study. One of the samples corresponded to the nerve being in association with a muscle variation found during the study.

All of the samples were fixed in 4% formaldehyde and processed until paraffin blocks were obtained. Then, they were cut into 4 µm slices and subjected to haematoxylin-eosin and Masson’s trichrome staining for histological analysis. The samples were analysed by the 3DHISTECH Pannoramic 1000 scanner, and images were constructed with CaseViewer 2.4 for Windows.

### 2.4. Statistical Study

All data were processed with Excel and SSPS 26. Qualitative variables were summarised as proportions. The concordance index (interclass correlation coefficient (ICC)) of the different measurements (nerve perimeter and diameter) was calculated for each anatomical point studied. If optimal ICC values were obtained, they were averaged and presented as the median or mean with their respective measures of dispersion.

The existence of statistically significant differences when comparing nerve diameters at the different anatomical points studied was determined using a median comparison test: Friedman’s test with Bonferroni correction. Moreover, for each of the points where significant differences were found, the theoretical perimeter of the circumference was calculated from the observed diameter. Later, it was compared with the real observed perimeter by Student’s *t*-test, with the aim of seeing if these measurements were comparable.

## 3. Results

### 3.1. Ultrasound and Anatomical Studies

The US study of the arm and forearm allowed us to identify the UN along its entire path in all the upper limbs. On the transverse scans, it appeared as an echogenic tubular structure containing hypoechogenic/anechoic discontinuous segments. On the longitudinal scans, it was identified as a homogeneous echogenic structure. This accuracy of the US technique enabled the placing of the intramuscular needles at specific points, as described in the methods.

The UN diverged from the brachial artery both posteriorly and medially and kept sliding on a self-contained membranous sheath at the medial aspect of the arm.

At the first anatomical landmark, the dissection study showed that the UN crossed the IMS to enter the posterior compartment and was accompanied by the superior ulnar collateral artery, which was identified in 100% of the upper limbs. The superior ulnar collateral artery was also identified in all the limbs in the US study as an anechoic oval structure (because it was injected by latex) emerging as a branch of the brachial artery.

The US transverse images showed the IMS to be a thin hyperechoic continuous line, while the longitudinal images showed it to be an echogenic continuous echoic band that interrupted the continuity of the nerve (Figure 2A). The US transverse images showed that the direction of the nerve became notably modified while crossing beneath the IMS, although the internal morphology of the nerve was preserved in most specimens. The dissection study revealed the UN as crossing the medial IMS (Figure 2B) that extended from the lower part of the crest of the lesser tubercle of the humerus to the medial epicondyle. Anatomically, the location of the intramuscular needles corresponded to the point at which the UN pierced this IMS in 91% of the upper limbs studied.

At the second anatomical landmark, 2 cm after the IMS, the UN was identified by US as resting on an echoic mass, which corresponded to the medial head of the triceps brachii, coinciding with the anatomical findings. The UN remained attached to the muscle belly by its epimysium in the posterior compartment (Figure 3). Before entering the osteofibrous tunnel in the elbow, the UN crossed an aponeurotic arch formed by the insertion of the brachial fascia into the medial epicondyle and olecranon.

In the elbow, when the UN pierced the osteofibrous tunnel, US images taken on the transverse axis showed it as a hyperechogenic structure covered by a well-differentiated hyperechogenic line that corresponded to the Osborne’s fascia (Figure 4A), as confirmed by the dissection study later, where it was seen as a fascial sheet (Figure 4B). These findings were objectified in 82% of the specimens both in the US and anatomical studies. However, in the remaining 18% of the specimens, which corresponded to two upper limbs from the same donor, US study showed a hypoechoic structure occupying the osteofibrous tunnel that reminded us of a muscular structure (Figure 5A). Accordingly, the UN remained adherent to the medial osseous wall (medial surface of the olecranon) and showed itself as a heterogeneous anechoic oval structure surrounded by a hyperechogenic area. The hypoechoic structure originated from the medial surface of the olecranon and inserted into the medial epicondyle, passing over the medial head of the FCU. We observed that this muscle had a second insertion point in the medial IMS, which corresponded to the insertion of the extension of the connective tissue covering the medial head of the triceps brachii. The dissection study revealed that this structure can be considered as an aberrant muscle mass (Figure 5B), and the UN was included in the epimysium of this muscular structure (Figure 5C). Moreover, we observed that the posterior branch of the inferior ulnar collateral artery accompanied the nerve through the osteofibrous tunnel, giving collateral branches to this muscle mass (Figure 6A). We also noted a nerve branch from the UN that innervated the FCU, providing collateral branches to innervate this muscular mass (Figure 6B). The presence of the aberrant muscle mass increased the compression of the nerve at this point, as the nerve could be anatomically seen to dilate in the lower third of the arm, proximally to the aberrant muscle mass. Moreover, the nerve in the upper limbs containing the aberrant muscle mass showed a larger diameter than those in the other upper limbs dissected. It should be noted that the UN recovered its fascicular morphology distally in the forearm.

Apart from these findings, when we studied signs of nerve compression in the elbow by US, we found that in 82% of the upper limbs studied, the nerve showed a focal homogeneous hypoechoic appearance, with loss of its normal internal fascicular pattern. In 55% of these specimens, the dissection study revealed feasible dilation of the nerve before its entrance into the osteofibrous tunnel. In one of them, the nerve was found rotated on its own axis once in the cubital tunnel of the elbow.

In the forearm, where the UN associates with the two heads of FCU (third anatomical landmark), the UN remained on a superficial plane that could be observed in the US (Figure 7A) and dissection studies (Figure 7B). In both studies, the UN was observed as a flattened nerve that recovered its oval structure distally. In the US study, the nerve was surrounded by a hyperechogenic area that, in the anatomical study, corresponded to the epineurium of the nerve anteriorly and to the extension of the internal aponeurosis of the FCU posteriorly.

The US study showed that at the third anatomical landmark, the UN preserved its internal fascicular structure in 63% of the upper limbs, while in 11% of the specimens, we had difficulties in identifying it and defining its morphology. The arcuate ligament of Osborne was well identified by US in only 18% of the specimens, appearing as a thin hyperechogenic line covering the UN. Anatomically, this ligament was observed as several fibrous bands that connected the two heads of the FCU to one another, forming an arch under which the UN moved distally in the forearm, which could then be used to locate the muscle bellies of the FCU and FDS at a deeper plane (Figure 7B).

These findings were different from those observed in the upper limbs containing the aberrant muscle mass, in which the lateral head of the FCU originated from the superficial surface of the olecranon and the deeper surface of the aberrant muscle epimysium instead of originating from the medial surface. We could see by US that this aberrant muscle mass covered the lateral head of the FCU proximally. Due to this anatomical arrangement of the muscles, the UN was mostly covered by the medial head of the FCU at its exit from the osteofibrous tunnel in the elbow.

As the UN travelled distally through the forearm, the US study showed that it associated with muscle bellies of the FCU medially and superficially and the FDS laterally and posteriorly (Figure 8A). Both muscles were identified as an oval echoic structure around the UN that simultaneously was surrounded by a hyperechogenic area. This area corresponded to the fourth anatomical landmark: the intermuscular aponeurosis between the FCU and FDS, as confirmed later in the dissection study. A detailed dissection revealed that this aponeurotic structure was composed of tendinous fibres from the FDS that originated from the aponeurosis of the FCU (Figure 8B). It was constant in all the specimens, although the intramuscular needle located at this point coincided in 82% of the upper limbs studied. The remaining 18% corresponded to the upper limbs containing the aberrant muscle mass, in which the structures surrounding the UN at this location were difficult to delimitate.

At this fourth anatomical landmark, the UN paved its way through the forearm, including the epimysium of the FDS as well as the FCU and its longus tendon reaching the pisiforme bone. These findings correlated with the ones obtained by US, although the exact point where the UN pierced the muscle epimysium was difficult to identify by US. 

Regarding signs of nerve compression in the forearm in the US study, the morphology of the UN was preserved in 82% of the upper limbs. In the remaining 18% of the specimens, the nerve was visualised as an anechoic oval structure. Some of the differences in the nerve diameter and perimeter observed between the different anatomical points in the anatomical study were corroborated by the statistical study.

### 3.2. Histological Study

A double-blind histological study was performed. Samples of the UN from the limbs containing the aberrant muscles mass were subjected to haematoxylin-eosin (Figure 9A) and Masson’s trichrome staining (Figure 9B) and were shown to present concentric fibrosis. These findings were not observed in the other samples analysed, which showed no significant or pathological changes (Figure 9C,D).

### 3.3. Statistical Results

As mentioned in the Material and Methods, we first analysed the concordance indexes (interclass correlation coefficient (ICC)) using the nerve diameter and perimeter values to determinate the level of consistency between them (Table 1). As a result, our variables were presented as the mean (standard deviation) or median (interquartile range).

To determine the normality of the distribution, the Shapiro–Wilk test was used (Table 2). To study the existence of differences between the diameter and perimeter measurements, a Friedman’s test with Bonferroni correction was applied (Table 3).

Figure 10 illustrates the anatomical points where we found statistically significant differences when comparing the nerve diameter and perimeter among the four anatomical landmarks. We set 0.05 as the level of significance. All the significant values shown in Figure 10 were adjusted with Bonferroni correction.

To compare the nerve diameter and perimeter measurements, we first estimated the nerve circumference using the nerve diameter measurements. As seen in Table 4 and Figure 11, there were no statistically significant differences between the observed nerve perimeter and its estimated circumference. Thus, although it is known that the nerve perimeter is more reliable for studying nerve dimensions, it is comparable to the nerve diameter and the nerve circumference estimated from it.

## 4. Discussion

Our results were in line with those of other studies showing that high-definition US can be used for a dynamic evaluation of UN integrity and its relationship with other anatomical structures in the arm and forearm [1,2,4,5,13,19,20,21]. Moreover, US can detect the presence of anatomical variations, which would explain the aetiology of nerve entrapment at specific anatomical locations. Therefore, US is a good tool for making good diagnosis as well as for planning appropriate treatments.

Previous US studies have reported that nerve pathological changes consist of a thinned nerve with proximal fusiform swelling and an increase in its hypoechogenicity, which entails a loss of its normal fascicular pattern [1,4,13,20,21]. It should be noted that an earlier study observed nerve enlargement distally to the compression site, although this is a rare finding [1]. US allowed us to identify these same changes specific anatomical points. US images taken along the longitudinal axis helped identify the critical points of compression by showing the nerve proximal and distal regions of the nerve from one exact point simultaneously with other methods, especially electrodiagnostic studies. Although changes in the external morphology of the nerve could not be observed, as no measurements of the cross-sectional area of the nerve were taken, given that it was not the aim of our study, we could discern changes in its fascicular morphology, which we interpreted as signs of compression. However, some studies have reported that the nerve can show a focal homogeneous hypoechoic appearance in normal subjects at sites where the nerve is subjected to chronic physiological compression, such as in the osteofibrous tunnel of the elbow [1].

In the dissection study, we observed that the UN diverged from the brachial neurovascular bundle in the anterior compartment, surrounded by its own sheath that was different from the one surrounding the other neurovascular structures. As the nerve moved distally, it pierced the IMS from the anterior compartment to enter the posterior compartment of the arm. This anatomical point correlates with what several authors describe as the arcade of Struthers, a fibrous canal consisting of the medial head of the triceps brachii and its aponeurotic expansion extending into the IMS as well as the internal brachial ligament described as a white, cord-like band [22,23,24,25]. However, the arcade of Struthers is a controversial structure, as some authors believe that it does not exist and is instead an anatomical variation of the medial IMS [7,23,26]. In accordance with those authors, and as we have described in this study, the IMS extends from the lower part of the crest of the lesser tubercle to the medial epicondyle, allowing the insertion of the triceps brachii and brachialis muscles. In our study, we also observed that the UN does not always cross directly beneath the IMS but can change its direction. Thus, these findings should be taken into consideration during surgical interventions.

In the posterior compartment, the UN was directly associated with the medial head of the triceps brachii being included in its epimysium. Some studies describe this nerve arrangement as a muscular channel formed of the triceps brachii through which the UN runs in the posterior compartment [26]. However, in some specimens, we noticed that the UN was directly associated with an extension of connective tissue that covered the nerve itself and the medial head of the triceps brachii as an aponeurosis while being attached to the whole length of the IMS. Anatomically, this tissue could also be defined as an extended membrane-like tendon. Despite this, it seems that the relationship between the UN and the triceps brachii protects the nerve from extrinsic compression, as there were no signs of nerve compression at this point. However, it should be noted that a hypertrophic muscle could explain nerve entrapment syndrome and/or nerve luxation in the elbow, as has been reported in some studies [17,27,28]. Conversely, we observed a direct association of the UN with the deep aponeurosis in triceps brachii muscle atrophy.

According to most studies, cubital tunnel syndrome involving the elbow and Guyon’s canal compression involving the wrist are the most common aetiology of ulnar neuropathy [4,5]. However, it should be pointed out that anatomical variations in the osteofibrous tunnel, such as the presence of accessory muscles, although infrequent, can be responsible for nerve entrapment.

Accessory muscles can be detected as incidental findings during routine US. Compared with MRI, US has the advantage of providing dynamic examinations that can show increases in muscle thickness and impingements on the UN, as we demonstrated with our upper limbs specimens [14].

Initially, when we identified the presence of an aberrant muscle mass occupying the osteofibrous tunnel in the US study, we interpreted it as the anconeus epitrochlearis muscle, which Testut and Latarjet described as a small muscle that extends transversally from the epitrochlear recess to the olecranon [6]. However, its dissection revealed the presence of an additional slip of the triceps brachii muscle, as it emerged from the lower part of the medial IMS and medial epicondyle, before attaching itself to the medial surface of the olecranon. This description coincides with the one provided by previous studies, which define it as a fourth head of the brachial triceps muscle [16]. There are a few reports in the literature referring to additional heads of the brachial triceps that have different origins such as the posterior aspect of the surgical neck of the humerus [27] or the proximal part of the humerus [18]. According to these studies, such additional slips may dynamically compress the UN at the level of distal humerus, consequently causing ulnar neuropathy around the elbow area. It could also explain the snapping triceps syndrome, which is described by some authors as a dynamic condition in which the muscle portion of the triceps dislocates the UN over the epicondyle, causing instability of the nerve [15,29]. Accordingly, our histological study confirmed fibrosis of the nerve that was associated with the presence of the aberrant muscle mass. This corroborates the importance of considering its existence in clinical practice. Furthermore, we described in detail the vascularity and innervation of this additional slip of the triceps brachii in the Section 3. Knowledge of these variations is very important in obtaining a precise diagnosis and treating UN neuropathy properly. Therefore, in the presence of these anatomical variations, a transposition of the nerve would not be the best surgical option, as it would worsen the symptoms.

Regarding the changes in nerve size, it should be pointed out the UN decreases when examined from the proximal to the distal end due to branches leaving the main nerve trunk. Likewise, the UN does not provide any collateral branch in the arm, while it supplies numerous branches in the elbow and forearm. Thus, size changes at this level might indicate signs of nerve entrapment. The statistical study of the nerve diameter and perimeter measurements confirmed that there were significant changes in the size of the UN in the posterior compartment of the arm and before and after its piercing of the osteofibrous tunnel in the elbow. However, when we compared the nerve diameter and perimeter before and after the crossing of the IMS, there were no statistically significant differences. As a result, we conclude that although the anatomy of this region and the path of the nerve in it increase the possibility of nerve compression at this point, the results from the statistical study do not prove this. Moreover, the histological study did not show pathological signs of compression in the posterior compartment.

However, we postulate that the upper third of the forearm is a critical area for the UN, as it is associated with several structures that can contribute to its compression, although most studies agree that UN entrapment occurs rarely in the upper forearm below the cubital tunnel [29]. Nevertheless, we observed that the UN becomes flattened when it pierces between the two heads of the FCU in 82% of the upper limbs dissected. Moreover, the statistical study confirmed significant differences in the diameter and perimeter between this point and a distal point in the forearm. Although it can be a potential site of compression, this enlargement at the exit of the nerve from the osteofibrous tunnel could be due to its disposition on a surface plane, as the histological study did not show pathological signs of compression in this area.

After associating with Osborne’s ligament and, straight after, the belly of the FCU, the UN pierced the intermuscular aponeurosis between the FCU and FDS. Some studies have reported that this aponeurotic structure could compress the UN below the cubital tunnel [29]. We did observe significant differences in the nerve diameter and perimeter before and after the crossing of this aponeurosis. However, the US study revealed no changes in the internal morphology of the nerve at this anatomical point.

As reported in some studies before, we noted that the nerve follows its path distally through the forearm below the FCU muscle and inside the epimysium of the FDS [30]. On the contrary, other studies report that the UN rests on the flexor digitorum profundus (FDP), remaining covered by its epimysium [6]. It was difficult to precisely observe these fascial structures by US, but it can help in the diagnosis by demarcating changes in the nerve.

## 5. Conclusions

US is a good imaging tool that can be used to make a complete diagnosis of ulnar nerve entrapment neuropathy, as it can reveal the path of the UN and its relationship with other anatomical structures, showing potential causes of nerve entrapment, such as the presence of additional muscles or other anatomical variations. Furthermore, this study demonstrated the existence of other potential anatomical locations of nerve compression apart from the most common ones. For this reason, it is important for surgeons and other specialists to know and be aware of them when assessing a patient in the consulting room or during the surgical procedure itself. Finally, future studies should analyse whether the area of the nerve measured by US is comparable with the area of the circumference calculated anatomically from measurements taken of the nerve diameter or perimeter. This could be used to verify the effectiveness of such a diagnostic test.

## 6. Limitations

The study had some limitations, as it was performed with a limited number of specimens. Thus, we cannot determine the real frequency of the anatomical variations. Another limitation was that we did not have the clinical history of the donors to confirm the clinical signs of nerve compression.

## Figures and Tables

**Figure 1 diagnostics-13-01332-f001:**
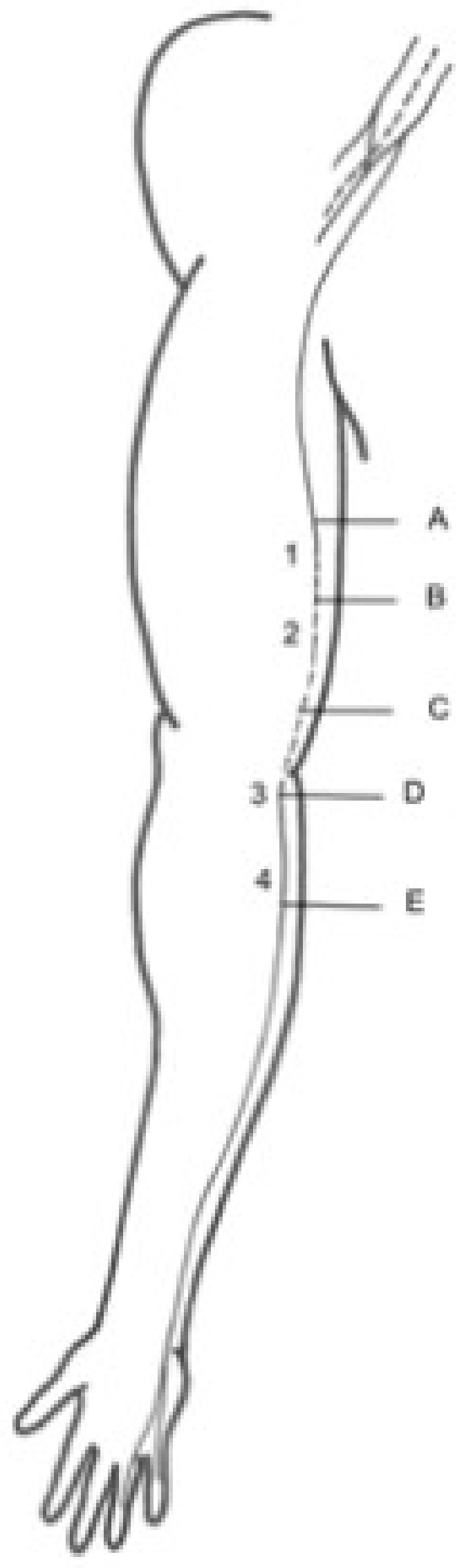
Numbers from 1 to 4 indicate the four specific locations of possible ulnar nerve entrapment to evaluate. Letters from A to E correspond to the anatomical landmarks where the nerve’s diameter and perimeter measurements were taken.

**Figure 2 diagnostics-13-01332-f002:**
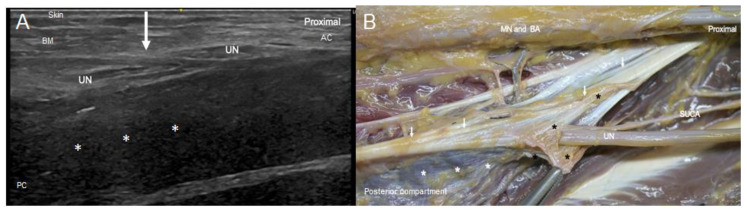
(**A**) US image in longitudinal axis of the ulnar nerve (UN) at the middle third of the arm. IMS is marked with a white arrow. The skin (SK), the anterior compartment (AC) with brachial muscle (BM), and the posterior compartment (PC) with the triceps brachii muscle (white asterisks) are indicated. The hipoechogenicity of this last muscle is due to the position of the probe to focus the UN and the septum. (**B**) Anatomical view of the UN crossing beneath the medial IMS (arrows), accompanied by the superior ulnar collateral artery (SUCA). We used a surgical clamp to pull the collagen fibres that make up the IMS (black asterisks). It is shown that, once in the posterior compartment, the UN remained covered by the medial head of triceps brachii muscle (white asterisks), which was inserted into the whole length of the IMS.

**Figure 3 diagnostics-13-01332-f003:**
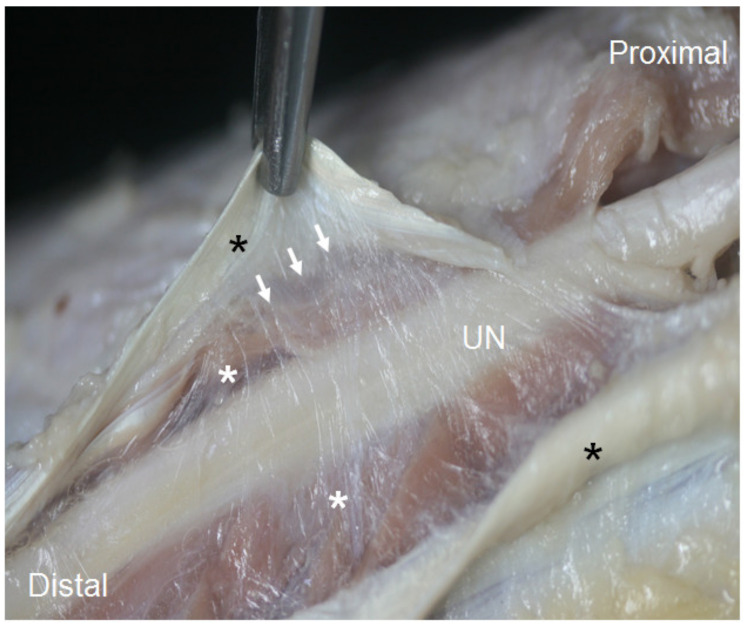
The UN in arm’s posterior compartment, after it crosses beneath the medial intermuscular septum. The white arrows mark the triceps brachii muscle’s epimysium in which the UN is involved, remaining attached to the medial head of triceps brachii (white asterisks). The black asterisks mark the membraniform tendon of this muscle, which has already been dissected.

**Figure 4 diagnostics-13-01332-f004:**
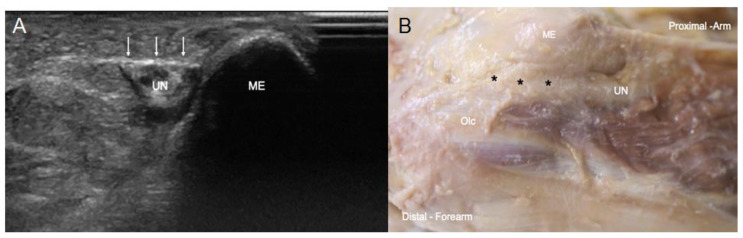
(**A**) US view of the UN at the osteofibrous tunnel covered by Osborne’s fascia (white arrows), laterally to the medial epicondyle (ME) (**B**) Anatomical view of the Osborne’s fascia marked with black asterisks and the UN passing between the olecranon (Olc) and the medial epicondyle (ME).

**Figure 5 diagnostics-13-01332-f005:**
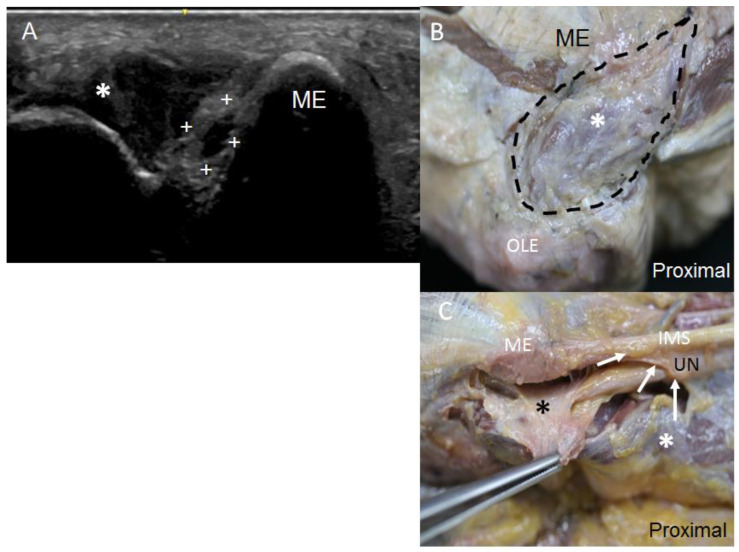
(**A**) Osteofibrous tunnel occupied by a muscular variant (white asterisk), which pressed the UN (white crosses) towards the medial wall of medial epicondyle (ME). It is shown that the UN appears as an oval anechoic structure, as it has lost its internal fascicular morphology. (**B**) Anatomical view of the muscular variant in the elbow delimited by intermittent black line. It is shown that it originated from the medial aspect of the olecranon (Olc) and is inserted into the medial epicondyle (ME). (**C**) The UN at the osteofibrous tunnel is included in the epimysium of this muscular variant (black asterisk), which was dissected from its multiple insertions. A white asterisk marks the medial head of triceps brachii. Finally, it is shown that the UN, before entering the osteofibrous tunnel of the elbow, crosses beneath an aponeurotic arch (white arrows) formed by the extension of connective tissue that covers the triceps brachii and is inserted into the intramuscular septum (IMS).

**Figure 6 diagnostics-13-01332-f006:**
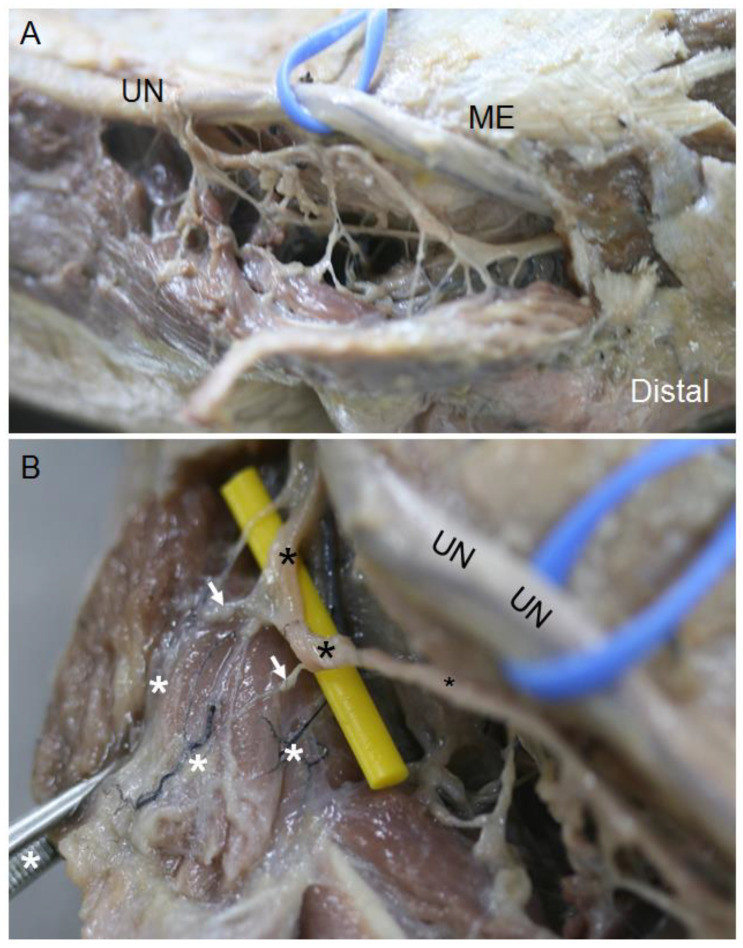
(**A**) Anatomical view of the ulnar nerve giving branches to the aberrant muscle at the level of the medial epicondyle (ME). (**B**) Collateral arteries (white asterisks), which originated in the posterior branch of the inferior ulnar collateral artery. At the very rear of the osteofibrous tunnel, the posterior ulnar recurrent artery accompanies the ulnar nerve (UN) at its path through the elbow. The white arrows indicate the nerve branches responsible for innervating this muscular variant, which originates in a UN branch (black asterisks).

**Figure 7 diagnostics-13-01332-f007:**
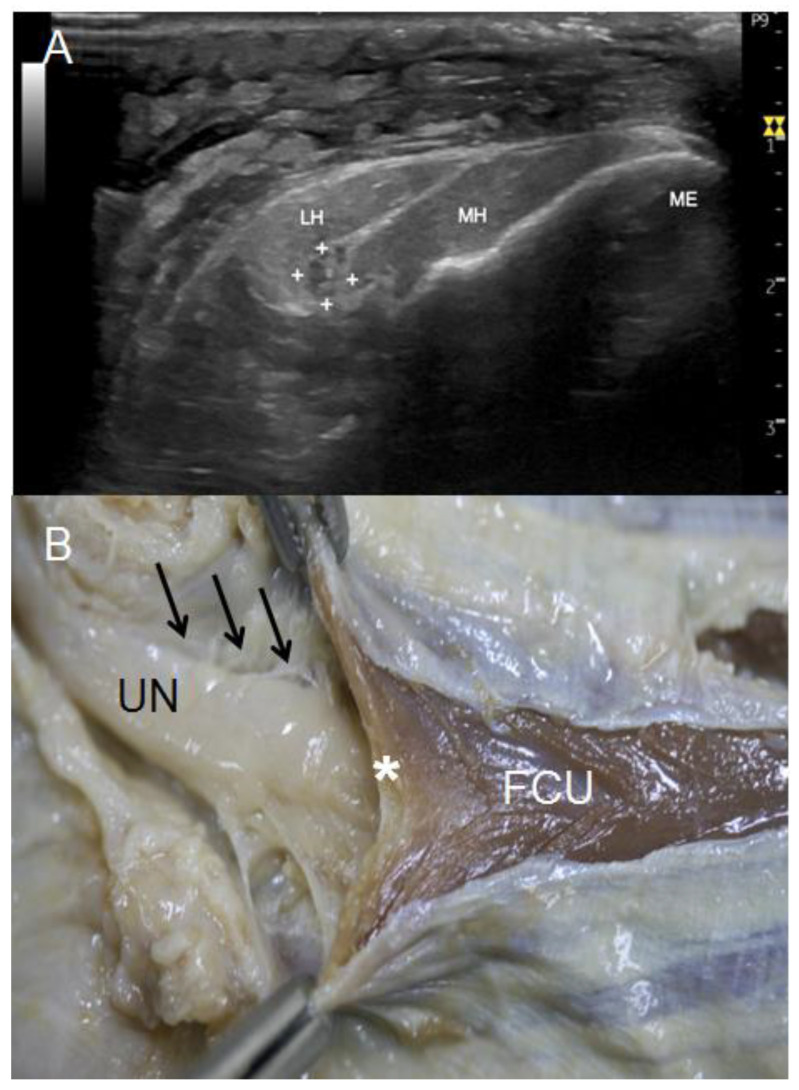
(**A**) US view of UN (white crosses) at its exit from the osteofibrous tunnel of the elbow, crossing between medial (MH) and lateral (LH) heads of flexor carpi ulnaris. (**B**) Anatomical view of the UN at this point. It is shown how it crosses beneath the Osborne’s ligament (white asterisk), which connects both heads of the flexor carpi ulnaris (FCU). In addition, note that the UN remains attached to the FCU’s epimysium (black arrows).

**Figure 8 diagnostics-13-01332-f008:**
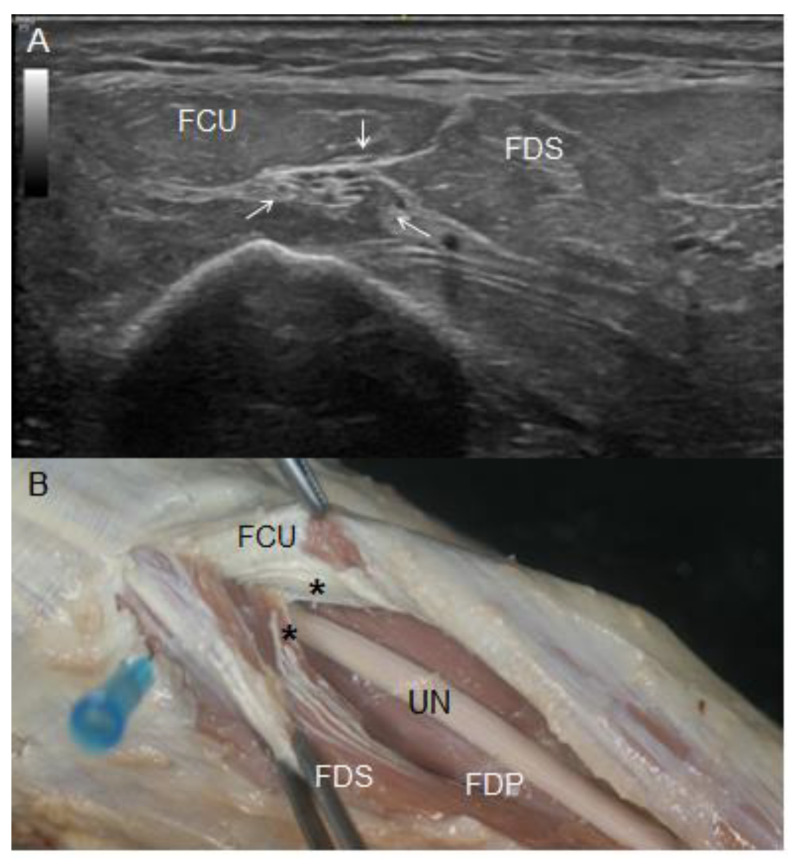
(**A**) Ultrasound view of the UN (white arrows) between the FCU and flexor digitorum superficialis (FDS) at the upper third of the forearm. (**B**) Anatomical view of the UN at this very point of the forearm. Note how it crossed beneath an aponeurotic structure (black asterisk), compressing the insertion of tendinous fibres of the FDS on the FCU muscle fibres. In this figure, UN was dissected from the FDS’s epimysium in which it was included, and it rests on FDP muscular mass.

**Figure 9 diagnostics-13-01332-f009:**
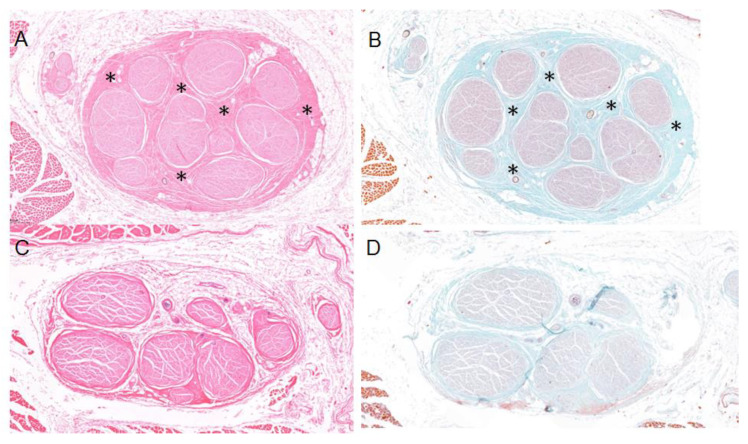
Histology samples of the UN in relation to the muscular variation and the fibrous tissue (black asterisks) can be observed between nerve fascicles (NF) as a sign of chronic extrinsic compression, stained with haematoxylin-eosin (**A**) and with Masson’s trichrome (**B**). Histology samples of the UN stained by haematoxylin-eosin (**C**) and Masson’s trichrome (**D**) without signs of fibrosis.

**Figure 10 diagnostics-13-01332-f010:**
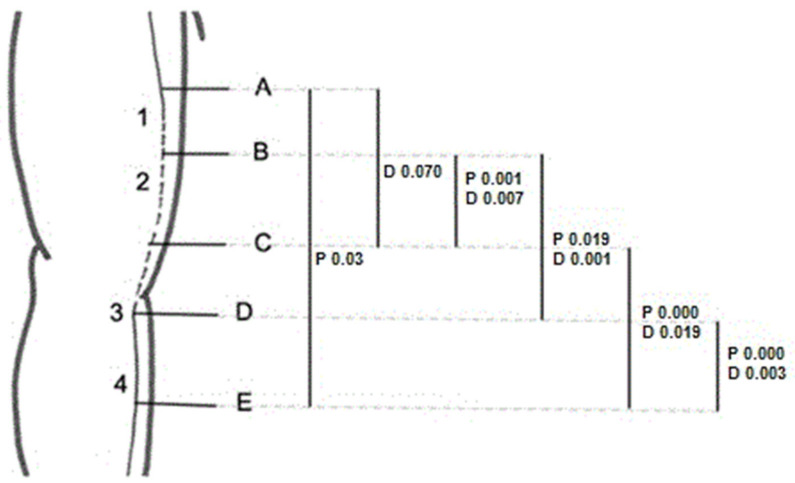
We found statistical significative differences when comparing nerve perimeter between A and E; nerve diameter between A and C; and nerve diameter and perimeter between B and C, B and D, C and E, and D and E.

**Figure 11 diagnostics-13-01332-f011:**
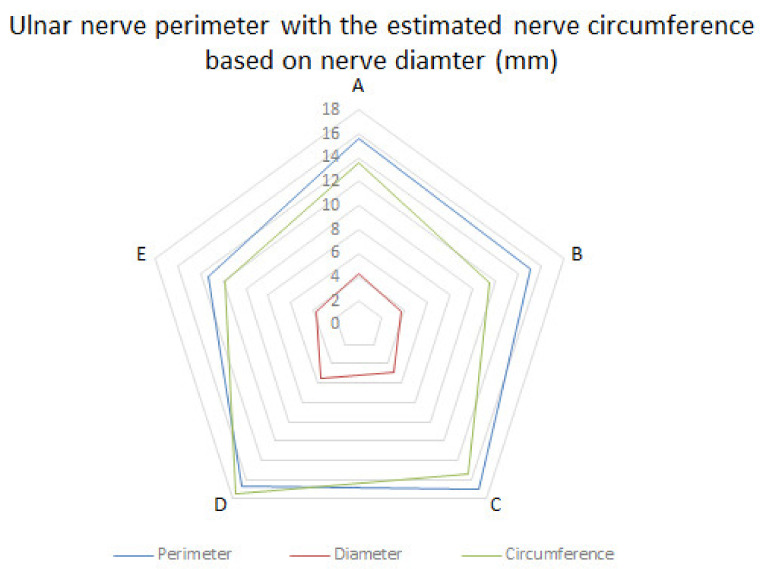
Ulnar nerve perimeter in comparison to the theoretical perimeter of the circumference calculated from the observed diameter.

**Table 1 diagnostics-13-01332-t001:** Diameter and Perimeter’s ICC in each anatomical landmarks studied.

Means	ICC	95% CI	F Test (*p*)
Perimeter	0.950	0.898–0.983	19.588 (0.000)
Diameter	0.642	0.250–0.885	2.795 (0.003)

Interpretation: <0.40 poor; 0.40–0.59 sufficient; 0.60–0.74 good; 0.75–1 excellent.

**Table 2 diagnostics-13-01332-t002:** Descriptive statistics.

Anatomical Landmarks	Perimeter				Diameter			
	Mean (SD)	IC 95%	Median (IQR)	Shapiro–Wilk (*p*)	Mean (DE)	IC 95%	Median (IQR)	Shapiro–Wilk (*p*)
**A**	15.54 (1.08)	14.85–16.22	15.65 (1.93)	0.254	4.25 (0.33)	4.03–4.48	4.33 (0.426)	0.055
**B**	14.83 (0.94)	14.23–15.43	15.01 (1.55)	0.873	3.94 (0.66)	3.50–4.38	3.66 (0.96)	0.473
**C**	16.94 (1.42)	16.04–17.84	16.95 (1.79)	0.481	5.26 (1.14)	4.5–6.02	4.93 (1.272)	0.097
**D**	16.73 (0.93)	16.14–17.32	16.73 (0.97)	0.641	5.46 (0.75)	4.96–5.97	5.58 (0.948)	0.636
**E**	13.52 (1)	12.88–15.16	13.27 (1.84)	0.257	3.83 (0.35)	3.6–4.06	3.75 (0.306)	0.535

**Table 3 diagnostics-13-01332-t003:** Statistical hypothesis testing.

Friedman Test	N Total	F	*p*
**Perimeter**	12	37.867	0.000
**Diameter**	11	25.673	0.000

**Table 4 diagnostics-13-01332-t004:** Comparison between nerve’s diameter and perimeter measurements.

	A	B	C	D	E	Mean (DS)	Difference	IC 95% Difference	t-Statistic (*p*)
**Perimeter**	15.65	15.01	16.95	16.73	13.27	15.522 (1.68)	−1.54	(3.73, 0.64)	−1.627 (0.142)
**Circumference**	13.60	11.50	15.49	17.53	11.78	13.98 (1.29)			
**Diameter**	4.33	3.66	4.93	5.58	3.75	4.45 (0.41)			

## Data Availability

The data presented in this study are available in the article.

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
