# Peer review of "Possible Points of Ulnar Nerve Entrapment in the Arm and Forearm: An Ultrasound, Anatomical, and Histological Study"

_diagnostics, 2023, doi:10.3390/diagnostics13071332_

Round 1
Reviewer 1 Report
-
The manuscript contains a description of two different studies. One is a cadaveric study and the other is a case report. These two studies should be separated and described in two different manuscripts.
-
The manuscript requires extensive English language proofreading and editing. Some parts of the text are difficult to understand.
-
The manuscript requires assessment by a statistical editor. The number of the studied specimens is too small to justify statistical analysis.
-
The information about the course of the ulnar nerve and the most common ulnar nerve compression sites in the upper limb is described three times – in the Introduction, Results and Discussion. This needs to be rewritten to avoid repeating the same information.
-
The Introduction section is too lengthy. It needs to be shortened.
-
„Some authors refer to this aponeurosis as a rest of the ulnar epitrochlear muscle of mammals, disappeared in man”. The sentence needs to be rewritten for a clear meaning, eg.” Some authors refer to this aponeurosis as a remnant of the ulnar epitrochlear muscle of mammals, no longer found in man”.
-
2.2 „Anatomical study”: The ulnar nerve arises from the „medial cord”, not the „medial fascicle”. Please, correct.
-
3.1 „Ultrasound and Anatomical Studies” (page 5, paragraph 4 from the top): Expressions „On one hand” and „On the other hand” are unfortunate and misleading when describing a condition affecting the upper limbs. Please, change.
-
Was there any history of compression neuropathy symptoms available in the files of the deceased limb donors?
-
The study's limitations should be listed, including the limited number of specimens.
Author Response
1. The manuscript contains a description of two different studies. One is a cadaveric study, and the other is a case report. These two studies should be separated and described in two different manuscripts.
We have eliminated the case report.
2. The manuscript requires extensive English language proofreading and editing. Some parts of the text are difficult to understand.
We would like to apologise. The text was reviewed by an English teacher but we have sent it again to the English correction service of the University.
3. The manuscript requires assessment by a statistical editor. The number of the studied specimens is too small to justify statistical analysis.
The simplicity of the model that this study presents dismiss the existence of bias that implicate as much it’s internal as it is external validity. The effect of a small sample, derivate of the difficulty of obtain specimens, only imply a lesser precision in the estimation of the population parameters but it doesn’t invalidate the results obtained in itself. Considering the datums obtained in the study, and based on the standard deviation observed, the number of cadavers used and the formula of the calculating sample for the quantitative variables, we were considering a mistake on the estimation that varies in general terms between 0,2 and 0,3 mm.
4. The information about the course of the ulnar nerve and the most common ulnar nerve compression sites in the upper limb is described three times – in the Introduction, Results and Discussion. This needs to be rewritten to avoid repeating the same information.
You are right, thank you very much for your observation. We have rewritten the different sections to avoid repeat the information
5. The Introduction section is too lengthy. It needs to be shortened.
Thank you very much, we have shorted the introduction.
6. Some authors refer to this aponeurosis as a rest of the ulnar epitrochlear muscle of mammals, disappeared in man”. The sentence needs to be rewritten for a clear meaning, eg.” Some authors refer to this aponeurosis as a remnant of the ulnar epitrochlear muscle of mammals, no longer found in man”.
Thank you very much for your observation. We have eliminated this sentence
7. 2.2 „Anatomical study”: The ulnar nerve arises from the „medial cord”, not the „medial fascicle”. Please, correct.
We are sorry for the anatomical mistake, we have corrected it.
8. 3.1 „Ultrasound and Anatomical Studies” (page 5, paragraph 4 from the top): Expressions „On one hand” and „On the other hand” are unfortunate and misleading when describing a condition affecting the upper limbs. Please, change.
Thank you very much for the observation. We have changed the expression. English corrector made a best text.
9. Was there any history of compression neuropathy symptoms available in the files of the deceased limb donors?
No, there is not. We have not the medical history of the donors, for this reason we found very interesting the histological presence of alterations at the ulnar nerve, in the donor with the aberrant muscle at the elbow.
10. The study's limitations should be listed, including the limited number of specimens.
Thank you very much for the observations. We have included them.
Reviewer 2 Report
Overall the paper is interesting and well organized. However, some minor points are reported below.
Lines 48-50: Please correct the sentence as follows: “Nonetheless, there could be other potential sites of ulnar nerve entrapment in the mid arm to distal elbow that should be considered in patients with ulnar neuropathy syndrome”
Lines 75-78: please rephrase these sentences.
Line 98: its all relations with other anatomical structures and anatomic variations; Please delete “all”
Lines 96-100: please rephrase these sentences.
Line 104: “Dissection Laboratory of the Faculty of Medicine and Health Sciences” of which University? Please add this information.
Line 107: “The axillary artery was injected”. Which substance was used? Please specify.
Lines 109-111: please correct “;” with “,”.
Figure legends are missing starting from Figure 2 up to Figure 10. Please add the figure legends and specify and explain in the figure legends all the symbols used in the panel of the figures (see for instance asterisks and arrows of Figure 4) as well as the abbreviation (see for instance ME, OLE of figure 5).
Figure 4B is not mentioned in the text. Please correct. The very same for figure 6, panel A and B.
Lines 318-322. Please revise this part since Figure 9A does not correspond to Masson’s trichrome staining. Please also revise this paragraph since it is not clear what the Autor means. The histological analysis requires more details.
The Authors did not use the journal’s template and some sections are missing, including “Author Contributions”, “Funding”, “Conflicts of Interest”, and “Data Availability Statement”.
Finally, it is important that the Authors add the following statements “Institutional Review Board Statement” and approval number and “Informed Consent Statement”, since any research article describing a study involving humans should contain this statement.
Author Response
1. Overall the paper is interesting and well organized. However, some minor points are reported below.
Thank you very much for your words.
2. Lines 48-50: Please correct the sentence as follows: “Nonetheless, there could be other potential sites of ulnar nerve entrapment in the mid arm to distal elbow that should be considered in patients with ulnar neuropathy syndrome”.
Thank you very much for your observation, however following the recommendations of the other reviewer we have suppressed this sentence.
3. Lines 75-78: please rephrase these sentences.
We have rephrased the sentences.
4. Line 98: its all relations with other anatomical structures and anatomic variations; Please delete “all”
We have deleted “all”.
5. Lines 96-100: please rephrase these sentences.
Thank you very much. We have rephrased the sentences.
6. Line 104: “Dissection Laboratory of the Faculty of Medicine and Health Sciences” of which University? Please add this information.
Thank you very much for the observation. We have not added it before because sometimes recommend not to included. Now it is included.
7. Line 107: “The axillary artery was injected”. Which substance was used? Please specify.
We have included black latex and the reference that we have used to injected the artery. Thank you very much for the observation.
8. Lines 109-111: please correct “;” with “,”.
We have corrected it.
9. Figure legends are missing starting from Figure 2 up to Figure 10.
We are very sorry. We thought that they were included. We have added.
10. Please add the figure legends and specify and explain in the figure legends all the symbols used in the panel of the figures (see for instance asterisks and arrows of Figure 4) as well as the abbreviation (see for instance ME, OLE of figure 5).
11. Figure 4B is not mentioned in the text. Please correct.
You are right, we are very sorry, we have corrected in the text and we have mentioned figure 4B in the text. Also, we have described the abbreviations that we used in the pictures. Thank you very much for your comments.
We have modified the figure 5 and we have annulled the figure 5B because we have eliminated from the article the case report following the recommendation of the other reviewer.
12. The very same for figure 6, panel A and B
Thank you very much for the observation. We are sorry. We have added the information in the main text and we have added the figure in the text.
13. Lines 318-322. Please revise this part since Figure 9A does not correspond to Masson’s trichrome staining.
Thank you very much you are right. We have added the staining of haematoxylin-eosin in the text for the fig 9A and corrected the Masson Please for the figure 9B.
14. Also revise this paragraph since it is not clear what the Autor means. The histological analysis requires more details.
We are sorry but we have not found this sentence in the text.
15. The Authors did not use the journal’s template and some sections are missing, including “Author Contributions”, “Funding”, “Conflicts of Interest”, and “Data Availability Statement”.
Finally, it is important that the Authors add the following statements “Institutional Review Board Statement” and approval number and “Informed Consent Statement”, since any research article describing a study involving humans should contain this statement.
We have added all this information. Thank you very much. We are very sorry.
Round 2
Reviewer 1 Report
I find the corrections introduced by the Authors mostly satisfactory.
I have a few minor suggestions:
-
Line 47: Please, change to „area innervated by the affected nerve”.
-
Line 55: The word „locations” can be omitted.
-
Figure 1 caption: „locations of possible ulnar nerve entrapment” should be written.
-
Figure 8 caption: Please, stick to the same abbreviation throughout the text. In the last line of the caption, the abbreviations FSD and FPD are used instead of FDS and FDP.
-
Lines 281-282: Please, restore the deleted information that both upper limbs in which the fourth head of triceps brachii was found, came from the same donor.
-
Discussion: It should be briefly mentioned within the main body of the manuscript that the two most common sites of ulnar nerve entrapment are the cubital tunnel and Guyon's canal (the way it is mentioned in the Abstract).
-
Discussion: When discussing the role of ultrasound in the diagnostics of compression neuropathies, other methods, especially electrodiagnostic studies should be mentioned briefly.
-
Line 583: The expression „as we have reported initially” can be omitted as the part of the manuscript that it refers to has been deleted.
Author Response
1. Line 47: Please, change to „area innervated by the affected nerve”.
We have change it. Thank you (Line 44).
2. Line 55: The word „locations” can be omitted.
We have not omitted following your advice of point 7 where we have added: the cubital tunnel of the elbow and the Guyon’s canal of the wrist (Line 50).
3. Figure 1 caption: „locations of possible ulnar nerve entrapment” should be written.
Thank you very much, we have added. (Line 142).
4. Figure 8 caption: Please, stick to the same abbreviation throughout the text. In the last line of the caption, the abbreviations FSD and FPD are used instead of FDS and FDP.
I am sorry for the mistake, you are right. We have changed them (Line 310).
5. Lines 281-282: Please, restore the deleted information that both upper limbs in which the fourth head of triceps brachii was found, came from the same donor.
We have restored that the fourth head of the triceps brachii is from the same donor (line 221).
6. Discussion: It should be briefly mentioned within the main body of the manuscript that the two most common sites of ulnar nerve entrapment are the cubital tunnel and Guyon's canal (the way it is mentioned in the Abstract).
Thank you very much, we have added in the introduction (line 50) and the discussion (line 412).
7. Discussion: When discussing the role of ultrasound in the diagnostics of compression neuropathies, other methods, especially electrodiagnostic studies should be mentioned briefly.
Thank you very much, we have added the sentence. (Line 375).
8. Line 583: The expression „as we have reported initially” can be omitted as the part of the manuscript that it refers to has been deleted.
Thank you very much for your observation. We have eliminated it (Line 445).